# Patient-Specific Guides for Accurate and Precise Positioning of Osseointegrated Implants in Transfemoral Amputations: A Proof-of-Concept In Vitro Study

**DOI:** 10.3390/medicina59030429

**Published:** 2023-02-22

**Authors:** Emir Benca, Beatrice Ferrante, Ewald Unger, Andreas Strassl, Lena Hirtler, Rickard Brånemark, Reinhard Windhager, Gerhard M. Hobusch

**Affiliations:** 1Department of Orthopedics and Trauma Surgery, Medical University of Vienna, 1090 Vienna, Austria; 2Center for Medical Physics and Biomedical Engineering, Medical University of Vienna, 1090 Vienna, Austria; 3Department of Biomedical Imaging and Image-Guided Therapy, Medical University of Vienna, 1090 Vienna, Austria; 4Division of Anatomy, Centre for Anatomy and Cell Biology, Medical University of Vienna, 1090 Vienna, Austria; 5Department of Orthopaedics, Gothenburg University, 40530 Gothenburg, Sweden; 6Biomechatronics Group, Massachusetts Institute of Technology, Cambridge, MA 02139, USA

**Keywords:** amputation, femur, osseointegrated implant, alignment, position, accuracy, precision, additive manufacturing, positioning guides

## Abstract

*Background and Objectives*: The treatment of transfemoral amputees using osseointegrated implants for prosthetic anchorage requires accurate implant positioning when using threaded bone-anchoring implants due to the curvature of the femur and the risk of cortical penetration in misaligned implants. This study investigated the accuracy and precision in implant positioning using additively manufactured case-specific positioning guides. *Materials and Methods*: The geometry and density distribution of twenty anatomic specimens of human femora were assessed in quantitative computed tomography (QCT) scanning. The imaging series were used to create digital 3D specimen models, preoperatively plan the optimal implant position and manufacture specimen-specific positioning guides. Following the surgical bone preparation and insertion of the fixture (threaded bone-anchoring element) (OPRA; Integrum AB, Mölndal, Sweden), a second QCT imaging series and 3D model design were conducted to assess the operatively achieved implant position. The 3D models were registered and the deviations of the intraoperatively achieved implant position from the preoperatively planned implant position were analyzed as follows. The achieved, compared to the planned implant position, was presented as resulting mean hip abduction or adduction (A/A) and extension or flexion (E/F) and mean implant axis offset in medial or lateral (M/L) and anterior or posterior (A/P) direction measured at the most distal implant axis point. *Results*: The achieved implant position deviated from the preoperative plan by 0.33 ± 0.33° (A/A) and 0.68 ± 0.66° (E/F) and 0.62 ± 0.55 mm (M/L) and 0.68 ± 0.56 mm (A/P), respectively. *Conclusions*: Using case-specific guides, it was feasible to achieve not only accurate but also precise positioning of the implants compared to the preoperative plan. Thus, their design and application in the clinical routine should be considered, especially in absence of viable alternatives.

## 1. Introduction

Amputations are among the most common surgical procedures worldwide, due to rising vascular and metabolic diseases, ongoing military conflicts and the occasional failure of limb-sparing reconstructions. Age-adjusted rates of lower extremity amputations range from 16.9 to 22.9 amputees per 100,000 population/year in different countries [1].

The interface between the residual limb and socket prosthesis has always been the weak point in the supply of amputees [2]. Thus, the direct anchoring of a prosthesis in the bone with an implant has proven an excellent alternative to the socket, especially for patients who suffer from complications such as skin rashes or infections, sweating, pressure points or insufficient soft tissue coverage [3]. A recent review concluded that due to osseoperception—the ability to directly perceive pressure, position and balance of the leg—patients with osseointegrated prostheses showed improved mobility, prosthetic function, and quality of life compared to socket prosthesis users [4]. The stability of the implant depends on its design and includes several variables, i.e., length, diameter, shaft or thread design, surface roughness, and finally the bone in which the anchoring takes place [5].

Among the most prominent bone-anchored implant systems is the Osseointegrated Prostheses for the Rehabilitation of Amputees (OPRA) (Integrum AB, Mölndal, Sweden). Unlike other systems on the market, the OPRA system’s anchoring system, the fixture, is a threaded cylinder and as such it provides not only a form-fitting but also a force-fitting connection to the bone. Therefore, the surgical procedure is fundamentally different as well. 

OPRA requires opening of the intramedullary canal, incremental drilling, thread tapping and fixture insertion. Most critical to this procedure is the correct intramedullary positioning of the fixture to avoid thinning of the cortical bone due to excessive drilling and underestimated anterior curvature of the femur and to minimise the risk of periprosthetic fracture. To achieve a desired fixture position, the surgeon is required to guide the drills by hand. Thus, the necessary force increases in sites with higher bone curvature and the position requires frequent radiographic control. Generally, there are currently two potential methods to facilitate an intraoperative aligning of surgical instruments, defining the final implant position: robot-assisted surgery and patient-specific instrumentation (PSI). While robot-assisted surgery was proven to improve the accuracy of implant positioning, e.g., in total hip arthroplasty [6,7,8] and bone tumor resections [9], its application is opposed by its high costs, steep learning curve, and low time efficiency. Thus, robot-assisted surgery would be less suitable for occasional surgical interventions, such as the insertion of osseointegrated implants opposed to more common procedures, such as total joint replacements. PSI is generated from three-dimensional (3D) imaging data, usually from computed tomography (CT) or magnetic resonance imaging (MRI). Based on the 3D representation of the structures of interest, instruments, matching the patient’s anatomy can be designed to facilitate intraoperative steps, such as bone resection [10] or implant positioning [11] and consequently improve the treatment outcome, e.g., achieve a more accurate implant alignment. Given its simple design, PSI could allow for accurate (low mean deviation from the planned position) and precise (low SD of deviations from the planned positions) positioning of the osseointegrated implants in limb amputees and potentially lower the operating time and radiation dose. This would allow not only for higher cost-effectiveness but also for lower complication and revision rates in these patients.

Therefore, this study aimed to investigate the application feasibility of additively manufactured (3D-printed) case-specific positioning guides for bone preparation and ossseointegrated implant insertion in the treatment of transfemoral amputees.

## 2. Materials and Methods

QCT scans of twenty femoral specimens were used to create digital 3D bone models, preoperatively define the optimal position of the fixture (intramedullary component of the OPRA implant system) and generate specimen-specific positioning guides. The guides were then additively manufactured. Following surgery, a second QCT series was assessed and corresponding digital 3D models were created. Corresponding digital 3D model pairs (pre- and postoperative) were digitally registered and the deviations of the operatively achieved from the preoperatively planned fixture position were assessed to evaluate the accuracy and precision of the positioning guides (Figure 1).

### 2.1. Specimens

A total of 20 specimens of human femoral specimens (10 female and 10 male or 9 left and 11 right), obtained from 12 donors to the Centre for Anatomy and Cell Biology, Medical University of Vienna were used (Table 1). The specimens belonged to a larger population that was used to assess the thermal effects during bone preparation and insertion of osseointegrated transfemoral implants [12]. The donors had provided written consent for their bodies to be used for research and education. The study was approved by the Ethics Committee of the Medical University of Vienna (1876/2019). Specimen characteristics are presented in Table 1. To prevent any change in the mechanical properties of the bone tissue, the specimens were fresh-frozen at −20 °C and exposed to room temperature 12 h before preparation. Femora were preselected based on the diameter of the bone and its medullary cavity to fit the 18.0 × 80 mm BioHelix fixture. Specimens were inspected for fracture, presence of lesions and prosthetic restoration. According to the previously set criteria, all specimens proved valid for the study. Finally, the femora were amputated in the middle third, respecting a minimum amputation height of 160 mm. The middle third of the femur features the highest curvature [13]. Thus, it is a challenging site to align the drills during surgery. The amputation height was chosen arbitrarily, within the middle third, to better reflect the clinical reality and documented (see Table 1).

### 2.2. Radiological Imaging

Radiological imaging included preoperative DEXA and pre- and postoperative QCT scanning. The Horizon DXA system (Hologic, Inc., Marlborough, MA, USA) was used to assess the areal bone mineral density in the proximal femur. The SOMATOM Force 196-slice dual-source CT scanner (Siemens Healthineers, Forchheim, Germany) was used to assess the 3D geometry as well as the volumetric density distribution and allow for the designing of specimen-specific positioning guides (see Section 2.3). The following parameters were used in QCT scanning: Dual-energy scanning mode, tube A and tube B: voltage: 150 and 100 kV, intensity: 220 and 460 mAs, respectively. Collimation was 196 × 0.6 mm, rotation time 500 ms and pitch-factor 0.5. QCT imaging sequences were reconstructed at a slice thickness of 0.6 mm and position increment of 0.4 mm using the Qr69 kernel and an iterative reconstruction mode (ADMIRE) at strength 3. QCT scanning was performed in the described manner, pre- and postoperatively, after metal removal (see Section 2.4).

### 2.3. Image Processing, Guide Design and Manufacturing

Cortical thickness (Ct.Th.) was measured at 12 locations (6 on medial and 6 on lateral side) along the implant axis in the coronal plane and reported as the mean value for each specimen. Images were further postprocessed using Mimics Research 21.0 (Materialise NV, Leuven, Belgium) and the global threshold segmentation for bone CT defining any voxel within the 226 and 3071 HU (Hounsfield units) range as bone. A 3D part was generated (setting optimal) for each specimen and postprocessed with the “wrap” (one pixel smallest detail, half a pixel as gap closing distance) tool. No further finishing steps were performed on the model to preserve its outer contour.

3-Matic Research 13.0 (Materialise NV, Leuven, Belgium) was used to design the specimen-specific positioning guides and experimental setup. The complete setup has been described previously [12]. Its most relevant parts are illustrated in Figure 2. In short, the designed positioning guides contained the specific negative impression of the 3D model of each femoral specimen. The impression was designed with an offset of 0.3 mm, accounting for underestimated bone tissue due to the partial volume effect (PVE) in the QCT imaging series and residual soft tissue. had a total length of 100 mm and an outer diameter of 40.0 mm. Each cylindric guide was designed to be concentric with the 18.0 mm diameter fixture within the medullary cavity. To achieve an optimal fixture position, a preoperative plan, considering the anatomy of the medullary cavity as well as the outer specimen contour, was designed. Not only was it necessary to center the fixture within the medullary cavity, but a minimum of 2.0 mm bone cortex also had to be maintained to achieve maximum implant stability [14]. Once designed, each guide was divided into two parts to facilitate the positioning along the specimens’ medial and lateral contours covering a total of two-thirds (67%) of the specimens’ circumference. Using a designed general positioning guide template which required only the adaptation of its inner surface to match the specimen-specific contour, it was possible to create the 3D model of the specimen and design the specific guide in less than 1.5 h.

The polyjet 3D-printer Connex 3 Objet 500 (Stratasys Ltd., Eden Prairie, MN, USA) using VeroPureWhite as model material and SUP706B (both: Stratasys Ltd., Eden Prairie, MN, USA) as support material was used to manfactur the positioning guides. The used model material has similar material properties as the MED610, which is licensed for medical applications in humans.

Digital three-dimensional parts created from the pre- and postoperatively acquired imaging series were imported into the 3-Matic Research software and registered manually. Automated registration of the two parts was less accurate due to the changed geometry in the postoperative series. Based on the preoperative plan, a cylinder was created that exactly matched the position and alignment of the fixture. The femoral shaft including the threaded implant canal was split from the proximal femur in the post-operative series. In the transverse plane, the implant-surrounding surface was selected. Based on this surface, a second cylinder, representing the position and alignment of the implant, was created. Ideally, the cylinder completely filled the threaded canal, which was controlled in the coronal and sagittal plane. If this was not the case, the workflow was repeated. Finally, the axes of both cylinders were projected in the coronal and sagittal planes. First, the offset including its direction (medial or lateral and anterior or posterior) between the two axes was measured starting from the most distal axis point of the preoperative implant canal. Second, if the axes intersected, the angle was noted under consideration of its tilting direction, distinguishing between abduction or adduction and extension or flexion.

### 2.4. Surgical Procedure

The surgical procedure has been described in detail previously [12]. In short, a single senior surgeon (GMH) performed all surgical steps, including drilling, thread tapping and fixture insertion. Drilling started with the largest drill diameter fitting the distal entry of the medullary cavity and continued with increasing sizes in 0.5 mm increments up to 16.5 mm. If the surgeon felt severe resistance caused by the presence of a large amount of cortex or accumulation of debris (bone chips), the instruments were withdrawn, cleaned, and reinserted. In the next step, an 18 mm thread was tapped to allow for the insertion of the prosthesis anchoring element or fixture (18.0 × 80 mm BioHelix fixture; Integrum AB, Mölndal, Sweden). All steps were performed by hand. Following successful insertion, the fixture was removed and all specimens were subjected to QCT scanning.

### 2.5. Statistical Analysis

Main outcome variables were defined as the misalignment of the achieved implant axis from the preoperatively achieved implant axis causing either hip abduction or adduction and extension or flexion (°). Furthermore, the axis offset was assessed in the medial or lateral and anterior or posterior direction measured at the most distal axis point of the preoperatively planned implant position.

Given the nonnormal distribution of some of the variables (assessed using the Shapiro–Wilk test), the Wilcoxon paired signed-rank test was performed to analyze for statistically significant differences between abduction/adduction and extension/flexion angles and medial/lateral and anterior/posterior offset. The Pearson product–moment correlation coefficient was computed to investigate linear correlations between (i) densiometric variables, (ii) outcome variables and (iii) descriptive and outcome variables. All tests were performed using the absolute (non-negative) values for all variables. Statistical significance was set at the 95% confidence level. The analysis was performed using IBM SPSS Statistics 27 (IBM Corp., Armonk, New York, NY, USA). Values provided in the text below are mean and standard deviation (SD), minima and maxima (MIN–MAX) of absolute values.

## 3. Results

Perfect alignment for abduction/adduction and extension/flexion was achieved in seven (35%) and and five (25%) specimens, respectively. The preoperatively planned and operatively achieved implant axis did not superimpose in any specimen, but reached low minima (≤0.02 mm) in the coronal and sagittal plane. The resulting misalignments were in abduction (*n* = 18 (90%)) and extension (*n* = 17 (85%)) with an axis offset in medial (*n* = 13 (65%)) and anterior (*n* = 12 (60%)) direction.

The means of the abduction/adduction and extension/flexion angle as well as the medial/lateral and anterior/posterior axis offset stayed well below 1° and 1 mm, respectively. The deviations of the outcome variables from the intraoperatively planned fixture position are presented in Table 2 and displayed in Figure 3.

The operatively achieved fixture position for each specimen with respect to the intraoperatively planned position is further visualized in Figure 4.

No statistically significant differences were found between the misalignments in abduction/adduction and extension/flexion angle (*p* = 0.085) and between the medial/lateral and anterior/posterior axis offset (*p* = 0.881).

The correlation analyses revealed a single significant correlation, namely between BMD and Ct.Th. (r = 0.496, *p* = 0.026). All correlation results are presented in Table 3.

## 4. Discussion

This study aimed to investigate the applicability of additively manufactured bone-specific positioning guides for bone preparation and osseointegrated implant insertion in the treatment of transfemoral amputees. 

The measured misalignment of the bone-anchoring element, the fixture in extension/flexion was twice as high as in abduction/adduction in comparison to the preoperative plan. This was the result of the higher anterior curvature of the femur and bears a risk of cortical penetration or fixture angulation [15]. The amount of bone removed during drilling is strongly governed by the local bone curvature. The anterior curvature of the femur is most excessive in the distal, followed by the middle shaft and shows significant interracial [16] and intersex [15] differences. Therefore, accurate alignment in extension/flexion of the fixture is essential. Using the positioning guides, the fixture axis misalignment in extension/flexion could not be eliminated, but reduced to <1° in 75% of specimens. Thus, the designed guides allowed for highly accurate implant positioning. Additionally, the positioning was highly precise due to the low deviation of the data (<0.7° SD for misalignment and <0.6 mm SD for axis offset). The maximum misalignment was 2.62°. As part of prosthesis fitting, the presented deviation in the fixture position could be well neutralized through available exoprosthetic components. Thus, the achieved positional precision allows for optimal restoration of the limb’s mechanics. More importantly, the highest recorded misalignment along with the highest axis offset (1.57 mm) would result in a 5.2 mm offset of the proximal fixture edge (calculated by multiplying the length of the fixture with the tangents of the misalignment angle and adding the length of the offset) in comparison to the preoperatively planned position. Given the preoperatively assessed mean cortical thickness of 8.1 mm, the minimal required cortical thickness of 2 mm would be still respected for this worst intraoperative outcome. Considering the low mean values for the misalignments and axis offsets in the present study, the mean deviation between the planned and achieved position of the proximal fixture edge would remain <1 mm. This would be considered an excellent operative outcome without any intraoperative radiation exposure. 

Bone drilling was started with the largest drill diameter fitting the distal entry of the medullary cavity. The axis of this initial drill canal defines greatly the axis of each consecutive drill, as well as of the fixture. Clinically, it is not expected that the initial drill produces significant axis misalignments since it is associated with the least bone removal and resulting low mechanical resistance. The mechanical resistance is increased by the amount of removed bone, which, as discussed before, depends on the local bone curvature and increases with the drill diameter. Thus, the final fixture axis offset was more likely the result of the misalignment in extension/flexion and abduction/adduction, than of a potential imprecision of the positioning guides. The angular misalignment of the fixture can easily be corrected in both planes during prosthetic fitting, with the mean offset below 1 mm and a maximum value of 2.15 mm. However, a high fixture axis offset in combination with a moderate angular misalignment can result in intra- or postoperative cortical penetration.

Direct comparison of the presented data with the literature is only partially feasible, as the investigated applications and geometries strongly vary. Over 99% of surface points in part comparison analysis for postoperative vs. planned position of 12 corrective osteotomies of the upper extremity were suited in the range of −3 mm to 3 mm [17]. Using drill guides for positioning of guide wires in femoral neck stabilisation, the reported mean displacements were <4 mm and the mean angulations < 5°, compared to the pre-operative plan [18]. Drill guides for pedicle bore in the lumbar spine showed a mean deviation under 2 mm [19]. Presented data lie in the range of the aforementioned deviations.

This study was performed on a comparatively large number of human anatomic specimens under consideration of BMD and Ct.Th. Nevertheless, it has several limitations. First, the length of the positioning guides (100 mm) in their current shape is clinically not applicable. In surgery, the part of the bone that is planned for resection plus 1 cm of remaining bone, where the periosteum could be pushed off and later on refixed, could potentially be used to accommodate the guides. A larger contact surface to the bone could harm the periosteum. Yet, the present study did not aim to introduce a medical device set but investigate the feasibility of its potential for clinical application. A reduction in length and resulting reduced contact surface between bone and positioning guides could potentially allow for more relative motions between the components and result in a deviated fixture position. Using stiffer materials for the positioning guides, such as titanium, could compensate for the deformation and displacement of the guiding components. Moreover, additional parts for the positioning guides could be considered, i.e., parts anchored in the proximal femur and firmly connected to the guides for the duration of drilling, which would allow for lower acting forces and moments onto them and consequently, lower resulting, displacements. Second, the quality and thickness of cancellous bone in younger patients, who normally receive this type of treatment, is more robust than of that in fresh-frozen anatomic bone specimens from older donors available for in vitro studies. On the other hand, following an amputation, the bone mineral density, as well as the cross-sectional area of the total bone and its cortex, reduce significantly [20]. The authors took this limitation into account by considering the specific BMD and Ct.Th. in data analysis.

## 5. Conclusions

Accurate bone preparation and positioning of the threaded intramedullary component of the osseointegrated implant system for transfemoral amputees relies on preoperative radiological images and surgeon’s skills and experience. Using additively manufactured patient-specific instruments (PSI), it was feasible to achieve not only an accurate but also precise implant position according to the preoperative plan. The resulting misalignments were well within clinically acceptable limits, without the necessity for intra-operative imaging for positional verification. Thus, their design and application in clinical routine should be considered, especially in the absence of viable alternatives.

## Figures and Tables

**Figure 1 medicina-59-00429-f001:**
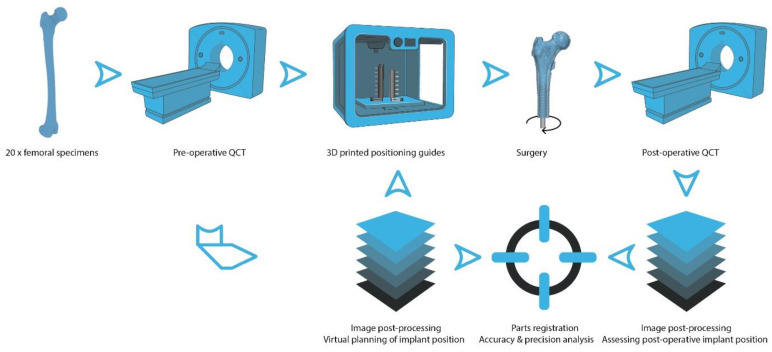
Graphical summary of the methodology: Twenty femoral anatomic specimens were scanned using a clinical QCT scanner. QCT images were postprocessed to create digital 3D bone models, define the optimal fixture (intramedullary component of the OPRA implant system) position and create specimen-specific positioning guides that were additively manufactured using a 3D printer. Following surgery, the specimens were QCT-scanned again and a postoperative digital 3D bone model of each specimen was created. In the last step, the 3D model pairs (pre- and postoperative) were registered and the deviation of the intraoperatively achieved implant position from the preoperatively planned implant position was analyzed. (Parts of the figure created with BioRender.com).

**Figure 2 medicina-59-00429-f002:**
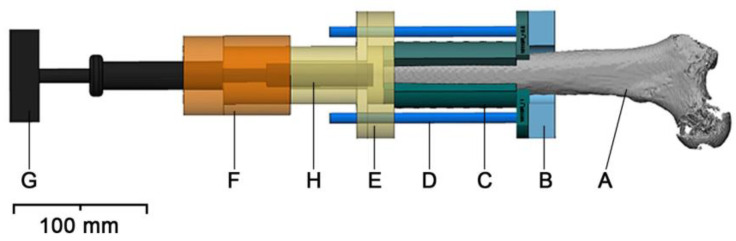
Experimental setup and positioning guides: The specimen (A) is mounted into a flange (B), attached to a table, with the aid of the positioning guides (C) and four steel 8 mm rods (D). The positioning guides accommodate one K-wire each. Two telescoping flanges, one (E) attached to the positioning guide and the steel rods and the other one (F) attached to the T-handle Hudson (G), allow specimen-specific alignment of the instruments and the fixture (H) insertion defined preoperatively.

**Figure 3 medicina-59-00429-f003:**
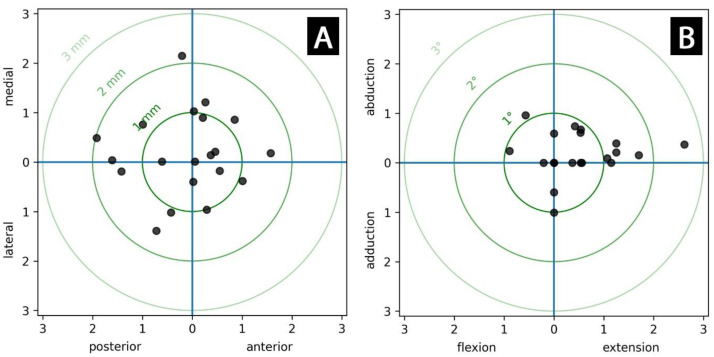
Scatter plots for deviation of (**A**) the axis (offset) (mm) and (**B**) the axis alignment (°) of the final fixture position compared to the preoperative plan.

**Figure 4 medicina-59-00429-f004:**
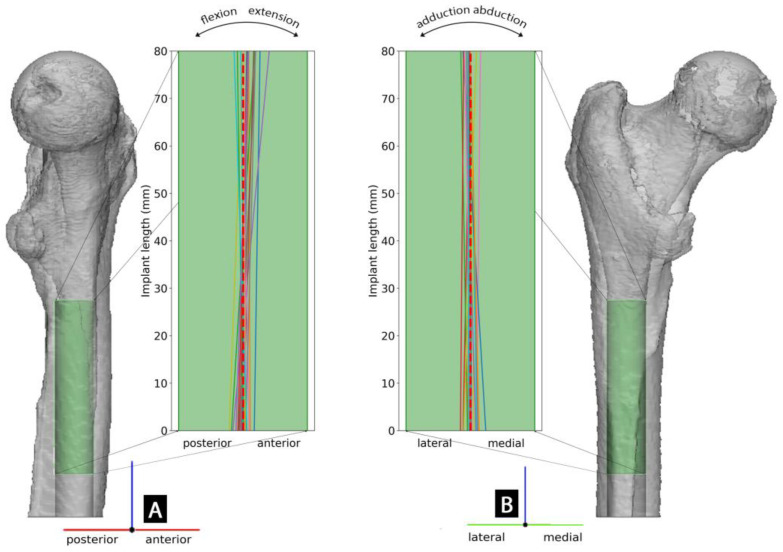
Visualization of the preoperatively planned fixture position (in a left femur) represented by the green cylinder in the 3D model and deviations of operatively achieved final fixture axis for each specimen in the corresponding 18.0 × 80 mm fixture projection in (**A**) lateral and (**B**) anterior-posterior view. The dashed red line represents the preoperatively defined fixture axis.

**Table 1 medicina-59-00429-t001:** Specimen data. Abbreviations: BMD (bone mineral density); Ct.Th. (cortical thickness).

Specimen	Donour No.	Sex	Age	Side	Length (mm)	BMD (g/cm^2^)	Ct.Th. (mm)
1	1	Male	87	Left	217	0.780	9.90
2	1	Male	87	Right	240	0.817	9.63
3	2	Male	76	Left	237	0.549	7.15
4	2	Male	76	Right	228	0.668	7.35
5	3	Male	72	Right	241	0.839	9.77
6	4	Female	65	Left	268	0.648	8.74
7	4	Female	65	Right	263	0.490	7.85
8	5	Female	91	Right	228	0.694	7.05
9	6	Male	78	Left	223	0.871	9.37
10	6	Female	101	Left	224	0.724	7.66
11	7	Male	74	Right	208	0.942	9.70
12	8	Female	75	Left	246	0.899	8.25
13	8	Female	75	Right	259	0.936	8.10
14	9	Male	86	Left	218	0.734	10.7
15	9	Male	86	Right	259	0.755	9.35
16	10	Female	95	Left	295	0.645	5.93
17	10	Female	95	Right	290	0.633	6.01
18	11	Male	67	Right	270	0.775	6.66
19	12	Female	69	Left	220	0.688	7.14
20	12	Female	69	Right	262	0.713	6.35
Mean ± SD (MIN–MAX)			77.4 ± 10.0 (65–101)		244.8 ± 24.3 (208–295)	0.740 ± 0.118 (0.490–0.942)	8.13 ± 1.40 (5.93–10.70)

**Table 2 medicina-59-00429-t002:** Deviation data for the achieved fixture alignment presented as resulting hip abduction or adduction and extension or flexion and axis offset in the medial or lateral and anterior or posterior direction measured at the most distal axis point of the preoperatively planned fixture position for all specimens. Values for abduction, extension and axis offset into medial and anterior direction are expressed as positive (+) and their mirrored quantities as negative (−) values in relation to the preoperatively planned implant position. Means, standard deviations (SD), minima (MIN), and maxima (MAX) were calculated for the absolute values.

Specimen	Abduction/Adduction (+/−) Misalignment (°)	M/L (+/−) Axis Offset (mm)	Extension/Flexion (+/−) Misalignment (°)	A/P (+/−) Axis Offset (mm)
1	0.15	0.18	1.70	1.57
2	0.00	−0.38	0.56	1.01
3	0.00	0.04	1.15	−1.61
4	0.00	−1.02	0.37	−0.43
5	0.37	−0.19	2.62	−1.42
6	0.09	0.01	1.07	0.05
7	−1.01	0.86	0.00	0.85
8	0.00	1.03	0.54	0.03
9	0.00	0.49	0.00	−1.92
10	0.00	0.90	−0.21	0.21
11	0.74	2.15	0.42	−0.21
12	0.00	1.21	0.00	0.26
13	0.96	−0.40	−0.57	0.02
14	0.67	−1.39	0.54	−0.72
15	0.61	0.76	0.53	−0.99
16	0.21	0.01	1.25	−0.61
17	0.59	0.21	0.00	0.46
18	0.39	0.14	1.25	0.37
19	−060	−0.96	0.00	0.29
20	0.24	−0.18	−0.89	0.55
	0.33 ± 0.33 (0.00–1.01)	0.62 ± 0.55 (0.01–2.15)	0.68 ± 0.66 (0.00–2.62)	0.68 ± 0.56 (0.02–1.92)

**Table 3 medicina-59-00429-t003:** Results of correlation analyses of interest, presented as the Pearson product–moment correlation coefficient with the corresponding *p*-value. Statistically significant correlations were highlighted with an asterisk (*).

	BMD	Ct.Th	Abd/Add	M/L Offset	Ext/Flex	A/P Offset
BMD	-					
Ct.Th	0.496 (0.026) *	-				
Abd/Add angle	0.022 (0.926)	0.098 (0.682)	-			
M/L offset	0.304 (0.192)	0.339 (0.143)	0.269 (0.252)	-		
Ext/Flex	0.064 (0.788)	0.172 (0.469)	−0.160 (0.501)	-	-	
A/P offset	−0.041 (0.865)	0.384 (0.095)	-	−0.342 (0.139)	0.344 (0.138)	-

## Data Availability

The data presented in this study are available on request from the corresponding author. The data are not publicly available due to national privacy regulations.

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
