# Peer review of "Patient-Specific Guides for Accurate and Precise Positioning of Osseointegrated Implants in Transfemoral Amputations: A Proof-of-Concept In Vitro Study"

_medicina, 2023, doi:10.3390/medicina59030429_

Round 1

Reviewer 1 Report

//

The article relates to a previous publication [3] by the authors describing a potential experimental positioning guide for bone surgical implants. It examines the feasibility of their proposed method to customise alignment of implants in transfemoral amputees. As a statistically- tested methodology devised by an international group of authors in complementary disciplines, the aim is to improve operative precision by prior insight into individual bone morphology. References cited indicate current interest in the topic which will be of special interest to anatomists and orthopaedic surgeons.

Title:  Would be more concise by deletion of “could allow” and “highly.”

Key words: “accuracy” and “precision” mean the same thing. What is meant by“additive manufacturing “……… maybe it could be defined in Line 23.

Abstract:  Would benefit from minor editorial attention (e.g., remove unnecessary adjectives).

Introduction: Provides the background context of amputation as amongst the most common, global surgical procedures; as an alternative to a socket prosthesis, an anchored implant offers better osseoperception, mobility and quality of life. Variables for consideration in implant stability include length, diameter, shaft or thread design, surface roughness and the nature of the bone in which it is anchored. The foremost system is OPRA with positioning by either robot- assisted surgery, or patient-specific instrumentation (PSI).

Materials and methods: Twenty femora (both sexes, left and right) were preselected and detailed in Fig 1 and Table 1. Pre- and post-imaging includes DEXA, CT, QCT with image sequence reconstruction (Mimics Research 21.0) detailed in Fig 2 and statistical tests (Wilcoxan paired rank test; Pearson product moment correlation) to analyse SD between a range of outcome measures.

Results: Deviation data for implant alignment Table 2, Scatter plots Fig 3, planned implant position Fig 4 and NS difference between alignments Table 3.

Discussion: Requires attention…….? Incomplete proposals…..?

Conclusions: A technological solution to a common anatomical surgical problem is evaluated and limitations discussed. For the uninitiated reader there are somewhat lengthy convoluted sentences making for a challenging read.

References:   These illustrate the current surgical interest, the technical advances, problem of femoral curvature, misalignment risks. Inconsistent use of capital letters in article titles…..sometimes upper case, sometimes lower case. Incomplete details for key reference [3].

Minor editorial points:

Line 23. What does “additively manufactured” mean?

Lines 31-36.

Line 26. ….imaging series WAS used.

Lines 31-36. A long, convoluted sentence , the meaning of which is not immediately clear to the average reader.

Lines 36-37. ………not only accurate but also precise………Repetition.

Line 40. Could be more concise.

Line 253-4.  “highly precise”, “highly accurate”………..unnecessary adjectives throughout devalue scientific currency.

Line 256. Sentence unfinished…….

Author Response

See file attached.

Reviewer 2 Report

To the authors:

This manuscript deals with the use of rapid prototyping technology to improve the surgical positioning of implants.

The authors have developed solid work, backed by a significant number of cases (20), but this work seems to fall short somehow; as a matter of fact, this could be an opportunity to improve the implants as well, or at least to propose new systems for the actual implantation surgery, not only for the positioning of the implants. In other words, the development of patient-specific alignment guides is interesting, but the potential of rapid prototyping approaches as this one is much wider.

Materials and methods starts with a graphical abstract, but this should be explained along this section; plus, the numbering is wrong (3.1 Specimens should be 2.1 instead). The average age of the donors is not presented, but the minimum is 65 years old; this should be commented by the authors, as it would affect the quality of the outputs, even having the BMD and Ct.Th values.

The amputation at a random shaft height would need to be explained as well, or at least the shaft height of the subjects and their percentage of amputation should be documented.

The protocol is said to be previously reported (ref #3), but more details are necessary for the understanding of the current manuscript. The surgical procedure was consistently conducted by a single surgeon, but it would be a major plus to have a larger cohort of surgeons, particularly with this number of cases.

The results need further clarification, as they are not easy to follow; the major outputs should have been detailed in the methods section, or at least some ground truth should have been presented for guidance.

The discussion is missing; this is a major flaw. Since discussion and conclusion are separated sections in this journal, the authors shall revise these sections. Furthermore, stating that additional components (e.g., exoprosthetic devices) could be necessary is contradictory to this approach, as this should resolve most of the misalignment problems. The amputation height is referred again, proving that it should have been consistently approached earlier in the study.

A real conclusion sentence is missing in the end.

Author Response

See file attached.

Round 2

Reviewer 2 Report

Most of my original comments are still to be properly addressed in this version of the manuscript; just to list a few points:

1- The authors have changed their conclusion to discussion, but the conclusion is just a single paragraph that doesn't reflect most of the work;

2- The graphical abstract is still not explained, and this is not suitable for initiating the methods section;

3- The previous study was not detailed and the number of references is still very limited.

Author Response

See file attached.
